# Attitude of Health Care Workers and Medical Students towards Vaccination against COVID-19

**DOI:** 10.3390/vaccines10040535

**Published:** 2022-03-29

**Authors:** Beata Jankowska-Polańska, Kathie Sarzyńska, Eddie Czwojdziński, Natalia Świątoniowska-Lonc, Krzysztof Dudek, Agnieszka Piwowar

**Affiliations:** 1Research and Innovation Center, 4th Military Teaching Hospital, 50-981 Wroclaw, Poland; bianko@poczta.onet.pl (B.J.-P.); natalia.swiat@o2.pl (N.Ś.-L.); 2Department of Internal Medicine Nursing, Faculty of Nursing and Obstetrics, Wroclaw Medical University, 50-996 Wroclaw, Poland; 3Department of Toxicology, Faculty of Pharmacy, Wroclaw Medical University, 50-556 Wroclaw, Poland; eczwojdzinski@gmail.com (E.C.); agnieszka.piwowar@umw.edu.pl (A.P.); 4Department of Logistics and Transport Systems, Faculty of Mechanical Engineering, Technical University of Wroclaw, 50-370 Wroclaw, Poland; krzysztof.dudek@pwr.edu.pl

**Keywords:** attitude, vaccination, COVID-19

## Abstract

The role of medical personnel in promoting vaccination and pro-health attitudes seems to be of key importance for protection against COVID-19. The aim of the study was to assess the attitudes of health care workers and students of medical faculties towards preventive vaccinations against COVID-19. A cross-sectional online self-administered survey was conducted among 497 people. The questions concerned attitudes towards vaccination as well as concerns about the side effects of the vaccine and contracting COVID-19. A positive attitude to vaccination was observed in 82% of the respondents. More than 54% respondents were concerned about side effects after COVID-19 vaccination. Medical students under 26 years had a more positive attitude towards COVID-19 vaccination, twice as high as among health care workers OR (95%Cl): 2.20 (1.03–4.66) vs. 4.06 (2.54–6.48), respectively. Students were more concerned than nurses about adverse effects of COVID-19 vaccine 3.8 (3.2–4.1) vs. 3.0 (2.7–3.5) and contracting the virus (1.7 (1.2–2.5) vs. 1.2 (1.0–2.0). Medical students had a more positive attitude toward vaccination than nursing students 4.2 (3.9–4.3) vs. 3.7 (3.3–4.3). In conclusion, predictors of positive attitudes toward vaccination were medical student status and young age.

## 1. Introduction

COVID-19 is a new disease caused by the coronavirus first detected in Wuhan, China, called SARS-CoV-2 [1]. The newly identified virus showed the ability to be easily transmitted and quickly spread to all continents causing a worldwide pandemic and becoming a global public health problem [2].

So far (January 2022) infection with the coronavirus causing the COVID-19 disease has been confirmed in 12% of Poland’s inhabitants, which corresponds to about 4.55 mln cases with a confirmed positive test for the presence of the SARS-CoV-2 virus. Since the beginning of the pandemic in Poland to January 2022, the number of deaths due to COVID-19 is 103,844.

At the beginning of 2021, new vaccines against COVID-19 began to appear successively. The first WHO-approved vaccine was COMIRANTY, manufactured by Pfizer and BioNtech. Currently (January 2022) the WHO has approved 10 COVID-19 vaccines [3]. The availability of various vaccines on the market is a key factor in containing the coronavirus pandemic, however, it should be emphasized that people hesitant to receive a vaccine against COVID-19 play an equally important part. Important in preventing the spread of SARS-CoV-2 is the identification of factors that influence attitudes toward vaccination. Research shows that one of these factors is people’s trust in health professionals. Therefore, the attitudes of health care workers can directly influence the behavior of the public [4]. Poland, along with Italy, Russia and France, is one of the countries in Europe with the smallest number of people willing to vaccinate [5]. Similar attitudes were found in the US, where acceptance rate for a potential COVID-19 vaccine was established at around 56.9% [6]. A few different elements should be taken into account when discussing reluctance towards vaccination, including fear of adverse effects, perceived low risk of contracting an illness or the illness’s low severity, lack of knowledge and distrust of conventional medicine and pharmaceutical companies [7].

Healthcare workers who carry out vaccinations, care for those infected with SARS-CoV-2 and educate patients in hospitals and clinics, have an impact on decisions regarding the implementation of mass vaccinations and the participation of citizens in preventive vaccinations. The behavior of Health Care Workers (HCWs) and medical students is a model for the rest of society. Constantly increasing awareness as well as building promotional potential among medical personnel who act as leaders and educators is crucial to the success of preventive pro-health actions [8,9].

Therefore, we decided to check the attitude of HCWs and medical students towards vaccination against COVID-19. It is important to determine what factors may influence the patient’s decision to participate or not participate in the mass vaccination program. In addition, the study researched the same subject among medical students who represent the future of the healthcare system, including its educational potential. Previous studies found that factors that may positively influence willingness to vaccinate against COVID-19: socioeconomic status, young age, female gender, perception of likely COVID-19 infection during a pandemic, and COVID-19 prosocial behaviors were facilitating factors [10].

The aim of the study was to assess the attitudes of HCWs and medical students towards preventive vaccinations and to determine what factors influence decisions regarding vaccination. In addition, a comparative analysis of the level of knowledge between HCWs, medical students and the impact of the respondents’ age was assessed.

## 2. Materials and Methods

The study used an original questionnaire, which was based on questions that were previously used in questionnaires assessing attitudes towards influenza vaccination [11,12]. To confirm the validity, clarity and the ease of administration of the planned survey, a pilot study was organized among 15 individuals of different ages and occupations (pre-study). After the required corrections, a final 32-question questionnaire assessing the knowledge, attitude and concerns regarding the COVID-19 vaccination was designed. The final version of the questionnaire consisted of four main parts: 1 (overall vaccination attitude) and 2 (positive vaccination attitude), where a higher score average indicated a more positive attitude towards vaccination. On the other hand, in parts 3 (concerns about the side effects of the COVID-19 vaccine) and 4 (fear of COVID-19), a higher average score corresponded with a less positive attitude towards vaccination against COVID-19. For each question, the respondents could receive a maximum of 5 points.

During the questionnaire development and pilot testing phase, none of the respondents made any comments regarding the design of the questions or problems with understanding the survey. The next step was to use the questionnaire to study the target population—health care workers and students.

The survey showed good psychometric properties. Cronbach’s alpha coefficient was 0.75 which means that the survey is a good research tool. Data were gathered in an anonymized fashion. The study received a positive opinion from Wroclaw Medical University’s Bioethics Committee No. 52/2021.

A cross-sectional online self-administered survey was conducted among 497 people. Participation was voluntary. The participants provided informed consent on the survey platform before they could proceed to the questionnaire. The criterion for inclusion in the study was work in a medical center or medical student status. The exclusion criterion was lack of consent to participate in the study.

The respondents were divided into two groups:Medical students (*n* = 328);Health care workers (*n* = 169).

The ratio of students to health care professionals was 2 times, which reflects the actual structure of the university hospital, where medical students are the most numerous group of people. The study was designed to reflect the environmental conditions in hospitals where the above ratios occur on a daily basis.

In the first part of the study, the responses in the entire surveyed group were compared, and in the next stages a comparative analysis was carried out, separating healthcare professionals from students of medical faculties.

The statistical analysis was performed using STATISTICA v. 13.3 (TIBCO Software Inc., Palo Alto, CA, USA). *p*-values less than 0.05 were considered statistically significant. For quantitative variables, basic descriptive statistics were calculated (Me—median, Q1—lower quartile, Q3—upper quartile, Min-minimum value, and Max—maximum value) and the compliance of their distributions with a theoretical normal distribution was checked using the Shapiro–Wilk’s W test. Comparisons were Performed with the Mann–Whitney U test for independent groups, or Kruskal–Wallis tests. Each categorical variable is presented as numbers and percentages. Spearman’s rho was used to estimate the correlation between the two variables. When comparing the attitudes of respondents towards vaccination, contingency tables and Chi-square tests of independence were used. The fractions in two groups were compared. It was assumed that the fraction differences will be significant when they amount to at least 20%. Assuming Type I error alpha = 0.05 and 1-beta test power = 0.80, the minimum size in each group should be *N*1 = *N*2 = 76.

## 3. Results

The response rate was 71% because 700 people were invited to participate in the study and 497 of them successfully completed (entirely) and returned the questionnaire. In the whole study group, the median (Q1–Q3) age was 24 (21–28), with the youngest participant being 18 and the oldest 64. The characteristics of the study group were as follows: every third respondent (35.0%) was a medicine student. The second largest group were people who had either completed or were in the course of nursing studies (22.3%). Less numerous groups were students of dentistry (17.7%) and pharmacy (12%). The remaining 12.9% were people declaring graduation or participation in the following courses: obstetrics and public health, dietetics, medical analysis. Among the employed, the largest number was nurses (21.5%), midwives (3.8%) with the least number of doctors (2.6%) and paramedics (0.4%). The results are presented in Table 1.

### 3.1. The Attitude of the Entire Study Group towards Vaccination against COVID-19

The results of the responses to individual questions showed that the respondents had a more positive attitude to vaccination against COVID-19 4.1 (3.6–4.3) than to vaccination in general 3.4 (2.9–3.9). On the other hand, they were also more concerned about vaccination adverse effects 3.6 (3.0–4.0). than they were about developing COVID-19 1.5 (1.0–2.3). (Table 2). The concerns expressed by the respondents were most often related to the possibility of contracting a disease through vaccination and the weakening of the immune system 1 (1–2). Respondents expressed concern about the mass introduction of vaccines and worried that the vaccine may be potentially dangerous and even ineffective. Among other concerns related to vaccination, there was the possibility of loss of physical and mental fitness 1 (1–2) as well as deteriorated professional and social performance 1 (1–1).

The attitude towards vaccination was generally dominated by worries about vaccination among patients with allergies and the chronically ill, followed by concerns about the rapid introduction of vaccines for the benefit of pharmaceutical companies. In terms of COVID-19 specifically, the respondents supported the validity of mass vaccinations 3 (2–3) and their health benefits 5 (4–5), though some respondents believed that the risk of falling ill is so low that vaccination on such a scale is not necessary 3 (2–4) (Table 2).

### 3.2. Attitude towards Vaccination against COVID-19 in the Studied Group of People by Occupation, University Degree and Age

A number of correlations were found between the results assessing attitudes towards vaccination against COVID-19 and the age, profession, gender, status and field of study of the respondents (Table 3). No statistically significant differences were observed in the attitude of the study participants based on gender. On the other hand, the medical student group had more positive attitudes toward vaccination than the HCWs group. Students had a significantly more positive attitude towards vaccination (domain 1) than nurses 3.6 (3.1–4.0) vs. 3.1 (2.6–3.4); *p* < 0.001). Midwives had a less positive attitude towards vaccination against COVID-19 (domain 2) than students 3.4 (3.0–3.8) vs. 4.1 (3.8–4.3); *p* < 0.001). Moreover, nurses had a less positive attitude towards vaccination against COVID-19 than students 3.7 (3.3–4.3) vs. 4.1 (3.8–4.3); *p* < 0.001). Students were more concerned about the adverse effects of the COVID-19 vaccine than nurses 3.8 (3.2–4.1) vs. 3.0 (2.7–3.5); *p* < 0.001 and contracting the virus 1.7 (1.2–2.5) vs. 1.2 (1.0–2.0); *p* = 0.023. Nursing students had a significantly less positive attitude towards vaccination (domain 1) than those studying pharmacy 3.1 (2.6–3.6) vs. 3.6 (2.9–4.0); *p* = 0.027 and medicine 3.1 (2.6–3.6) vs. 3.6 (3.3–4.0); *p* < 0.001. On the other hand, students at the faculty of medicine had a more positive attitude towards COVID-19 vaccination than those studying nursing 4.2 (3.9–4.3) vs. 3.7 (3.3–4.3); *p* < 0.001.

In an age-related analysis, people aged 55–64 showed less willingness to be vaccinated against COVID-19 (domain 2) compared to the group aged 18–24, median (Q1–Q3) 3.8 (3.3–3.9) vs. 4.2 (3.7–4.4); *p* < 0.001. The situation was slightly different in domain 3 (concerns about the effects of the COVID-19 vaccine), as participants aged 18–24 were more afraid of the effects of the vaccination compared to the 55–64 age group 3.7 (3.2–4.0) vs. 3.1 (2.4–3.4) *p* < 0.001.

### 3.3. Assessment of Attitudes towards Vaccination—Detailed Analysis

In the study group, only 1.0% (*n* = 5) of people had a strong negative attitude towards vaccination in general (domain 1), with some being rather negative 17.7% (*n* = 72), neutral 37.1% (*n* = 183), and most people, as much as 47.2% (*n* = 237) rather positive. There were no people with an overwhelmingly positive attitude towards vaccination. In domain 2 (positive attitude towards vaccination against COVID-19), most people had a rather positive (67.1%) or strongly positive (14.9%) attitude towards vaccination against COVID-19, while approximately 14% of respondents could not take a position on this matter, and the remaining few people expressed a negative attitude: either strongly (0.8%) and rather negative (3.2%) (Table 4).

In domain 3 (concerns about the adverse effects of the COVID-19 vaccine), the largest group was represented by people with concerns about the adverse effects of vaccination (50.2%), another 30.3% were people who did not have an opinion, only 1.2% of respondents showed a strong lack of concern. However, in domain 4, which represented the fear of falling ill with COVID-19, the distribution was as follows: strong lack of concern—44.2% (*n* = 220), lack of concern—33.5% (*n* = 168), did not know or had no opinion—17, 1% of the respondents (*n* = 83). A total of 4.4% (*n* = 22) of the respondents had concerns of getting sick, and only 0.8% (*n* = 4) (Table 4) had strong concerns.

### 3.4. Analysis of the Relationship of Age Variable with Individual Domains of the Questionnaire

The selected variables were analyzed on the basis of the results of the general attitude to vaccination studies. The study took into account: age and professional status.

Positive attitudes towards vaccination were considered to be average scores of 4 points or more (corresponding to „yes” and „definitely yes” answers) in the first, second and third domains. For domain 4 (fear of COVID-19), 3 points was adopted as the cut-off value. A statistically significant negative correlation was observed between the level of positive attitude to vaccinations (domine 1, 2 and 3) and the age of the respondents (Table 5).

The results of the ROC curve analysis (Figure 1) were used to determine the cut-off value for age. In the group of people under 26 years of age, the chance of a positive response to vaccination in domain one is five times higher than among older people (OR = 5.23). In the second domain it is almost four times greater (OR = 3.87) and in the third domain it is over four times greater (OR = 4.44, Table 6).

The chance of a positive attitude towards vaccination in the first and second do-mains in the group of students of the medical faculty is more than twice as high as among health care workers (ORadj. = 2.58 and 2.20, Table 7).

The chance of a positive attitude towards vaccination in the first and second domains in the group of students of the medical faculty is more than twice as high as among health care workers (ORadj. = 2.58 and 2.20).

In conclusion, the study showed many significant relationships between positive attitudes towards vaccination and individual health professions and age under 26. In addition, a group of medical students had better attitudes toward vaccination than health care workers. There were no significant differences between the level of knowledge about vaccinations and gender.

## 4. Discussion

Reluctance by parts of the public to accept the COVID-19 vaccine varies from country to country. The most common reasons for resistance towards vaccination are concerns about side effects or fear of the vaccination’s long-term impact that could deteriorate one’s health. A large part of the public expresses a desire to vaccinate, but at a later date, wanting to wait for reactions among those relatives and friends who already underwent the procedure. As reported by Grochowska et al. in a study conducted on a group of 419 Polish doctors, nurses, medical students, and other allied health professionals, safety and efficacy of COVID-19 vaccinations would persuade 86.3% of hesitant and those who would refuse to be vaccinated, while only 3.1% of all respondents claimed that no argument would convince them to get vaccinated [13]. In a study conducted by Della Polla et al., a lack of trust in influenza vaccine safety and efficacy was observed in more than half of the healthcare workers, though it did not have a statistically significant impact on practicing vaccination behaviours [14]. The public perception of mass vaccination may have been disrupted by the confusion prevailing in social media, which included references to conspiracy theories [15,16,17]. Poland is one of the countries with the lowest COVID-19 vaccine acceptance rates, which amounts to 56.3% of those willing to accept the vaccine [10]. Now (01/2022) it turns out that the estimates turned out to be accurate, because 57.1% of the Polish population is fully vaccinated against COVID-19 [18].

In our own research, 82% of people asked had a positive attitude towards vaccination against COVID-19, while another study showed that 92.1% of doctors were willing to accept this vaccine, while among pharmacists only 64.7% showed such willingness [19]. Babicki and Mastalerz-Migas assessed the changes in the attitudes toward vaccination against COVID-19 over time (before and after the vaccination program started) in Poland, showing only a slight increase in the willingness to get vaccinated against COVID-19 over the course of the study [20]. Additionally, people with a higher level of education and health care workers showed a more favored attitude toward vaccination. However, despite the passage of time and the increasing experience with the new types of vaccines against COVID-19, the percentage of people afraid of the potential complications after the vaccination has not decreased significantly and the concerns related to the ineffectiveness of vaccination have dramatically increased. Interestingly, in a study conducted by Grochowska et al., although most respondents—62.5% (262/419) indicated they had more trust in the influenza vaccine, more respondents intended to get vaccinated against COVID-19 in the 2020/2021 season [13]. This is in line with our own findings—in the study group it was shown that 47.2% of people had a positive attitude towards vaccination in general, while almost twice as many people declared a positive attitude towards vaccination against COVID-19. On the other hand, a study of 490 Italian HCWs showed that participants who were willing to receive the influenza vaccine were also more likely to receive COVID-19 vaccination, as well as showed less concern about influenza vaccine side effects [21]. Higher willingness to vaccinate among doctors and medical students fits well with the vaccination schedule, as they belong to the group of the highest priority for vaccination in Poland [18].

In turn, in the work of Kanyike et al. [22] on a group of medical students in Uganda, it was shown that the majority of the participants (*n* = 376, 62.7%) were not willing to be vaccinated against COVID-19. The most cited reasons for not taking up the vaccine were concerns about safety (*n* = 242, 64.4%) and having heard or read negative information about the vaccine (*n* = 201, 53.5%). Of those that reported having heard negative information about the COVID-19 vaccine (*n* = 575, 95.8%), the most frequently mentioned sources were social media (*n* = 521, 90.6%), and friends (*n* = 325, 56.5%). In our research, only 5.2% were afraid of the negative consequences of falling ill with COVID-19. Fear of vaccine side effects was similarly assessed, with only minor concerns identified. However, 32.14% of this research group indicated concern about the long-term effects of the COVID-19 vaccine. Students specifically were more concerned about the adverse effects of the COVID-19 vaccine compared to nurses, while remaining generally positive about the COVID-19 vaccine, with nursing students being the least positive and emergency medicine students the most positive. On the other hand, a study of 793 undergraduate nursing students from 12 Polish universities revealed that in the spring of 2021, 77.2% of study participants were already vaccinated against COVID-19 and approximately 50% of students in the unvaccinated group declared willingness to get a vaccination [23]. A study conducted by Talarek et al. in 2020 among the students of Warsaw Medical University showed that a majority (94.6%) of students expressed their intention to receive a hypothetical COVID-19 vaccine. Additionally, the study reported that older students (4th–6th year) were more often vaccinated against influenza and that a positive attitude towards influenza vaccination was linked to the intention to receive a COVID-19 vaccine [24].

Though compared to students in our study, doctors had a more positive attitude towards vaccination in general and vaccination against COVID-19, medical students were more willing to accept both vaccines in general and the COVID-19 vaccine than all the other HCWs in the study (nurses, midwives, emergency medicine workers). A total of 98.41% of physicians declared that in their lives they had taken compulsory vaccinations against other diseases. Another study showed physicians were more willing to be vaccinated against COVID-19 (94.44% vs. 91.99%), less concerned with vaccine side effects, as well as less frequently believed in conspiracy theories (3.17% vs. 8.69%) compared to medical students [20]. To compare, our research showed that students and physicians were equally concerned with the side effects of getting the COVID-19 vaccine and the possible outcomes of contracting COVID-19. In another study, nearly all participants had positive attitudes towards vaccines and agreed they would likely be exposed to COVID-19; however, only 53% indicated they would participate in a COVID-19 vaccine trial and 23% were unwilling to take a COVID-19 vaccine immediately upon FDA approval. Students willing to immediately take the vaccine were more likely to trust public health experts, have fewer concerns about side effects and agree with the vaccine mandate [25]. As reported by Lindner-Pawłowicz, doctors (64.7%) and medical students (63.7%) most often declared confidence in vaccines compared to nurses (34.5%). Distrust about vaccine safety was declared by nurses (46.6%) and pharmacists (40.0%) [26]. We have obtained similar results, with nurses and midwifes being the least positive towards vaccination.

The age of the participants and their attitude towards vaccination as well as the fear of possible side effects of the COVID-19 vaccine shows similar trends as the occupation analysis with younger study participants being more positive about vaccines, while at the same time displaying a considerably greater fear of vaccine side effects. In contrast to our findings, a study by Papagiannis et al. reports that older age of Greek healthcare workers was associated with the likelihood of COVID-19 vaccination acceptance [27], while a study of Canadian healthcare workers reveals that age over 50 is independently associated with vaccine acceptance [28]. On the other hand, Di Gennaro et al. found that predictors of vaccination acceptance included younger age [29]. The conflicting nature of these findings suggests age alone cannot be confidently treated as a single independent predictor of good attitude towards vaccination.

Interestingly, in our own study, it was shown that 47.2% of people had a positive attitude towards vaccination, while almost twice as many people declared a positive attitude towards vaccination against COVID-19. According to the study by Lazarus et al. [30], in the 19 countries with high levels of social trust in government (e.g., China, South Korea, Singapore), the approval rate of the COVID-19 vaccine exceeded 80%. This finding indicates that an increase in government’s reliability may have a beneficial influence on the attitude towards COVID-19 vaccination within the society. However, recommendations and adequate explanations provided by respected healthcare workers may also play an important role. Authors concluded that to obtain herd immunity, building increased trust among the general population is needed, and the elements that define and build trust must be understood with interventions crafted accordingly. Respondents from Poland reported the highest proportion of negative responses (182 of 666, 27.3%), which means that 72.7% of people were willing to undergo vaccination, which is not a big difference compared to 82% in our study, taking into account the occupational differences between study participants. Moreover, if an individual trusted their government, they were more likely to respond positively to their employer’s vaccination recommendation than those who did not trust the government [31]. This is also confirmed by the studies by Van Der Weerd et al. [32] where trust in the government positively influenced an intention to accept vaccination, but not to an intention to adopt protective measures (such as additional hygienic precautions). Similarly, in the studies by Jeżewska-Zychowicz et al. on a group of 1033 Poles, it was shown that trust in the government was average in this group. Among the many different sources of social trust, doctors fared the best [33]. As reported by Marinos et al., participants being informed about the COVID-19 vaccines by social media had lower COVID-19 vaccination coverage than Greek health workers being informed by other sources; receiving information on COVID-19 vaccination from the national public health authorities was also an independent factors of reported COVID-19 vaccination coverage [34]. The study conducted Haque at al. [8] reported that the demonstration of preventive behaviors increased with the level of education as well as the age of medical students and physicians. Generally, it was indicated that in public health crises such as the COVID- 19 pandemic, it is important to plan scientific knowledge-based public education; take initiatives in accordance with the cultural, social, economic, religious, and local characteristics of the societies; and conduct public health studies covering the whole society [35]. The current information system regarding passing the knowledge on the safety and efficiency of vaccination should be improved especially in medical groups above 26 years. Equipping medical personnel with personal and professional skills to better contribute to the healthcare system in the present pandemic and beyond to promote vaccination and increase health awareness of the society seems to be especially important.

## 5. Conclusions

The conducted study showed numerous dependencies between the willingness to receive the COVID-19 vaccine and the profession, student status and age. Compared to the rest of the study participants, students had a significantly more positive attitude towards vaccination compared to the rest of the respondents, which confirms the important role of medical students not only in future patient care, but especially in present prevention and promotion of pro-health behaviors. Other factor showing positive attitudes toward vaccination was young age. There were no correlations between attitudes toward vaccination and gender of the subjects.

## 6. Study Limitations

The study is limited by the relatively small group of respondents (*n* = 497). The number of people included in the study is lower than the originally assumed 700 people, but some participants withdrew from the study and some questionnaires were eliminated from the study due to either not signing the informed consent agreement or numerous missing answers to individual questions. Another limitation is the different numerical values of participants between students and healthcare professionals. People taking part in the study worked in different health institutions or studied at different universities, however, the study was conducted only in one region. The study used an original questionnaire due to the lack of standardized research tools assessing attitudes towards vaccination against COVID-19.

## 7. Practical Implications

We can suggest that COVID-19 vaccination campaigns should attend to raising awareness and the role of medical students and focus on promotion the vaccination against COVID-19. Minimizing the fear of vaccine side effects should be a part of this promotion. Another important stage of a potential social campaign is targeting and adjusting awareness activities to those areas of ignorance or fear that constitute the biggest problem in a given social group, preventing the implementation of mass vaccinations against COVID-19. Sharing information about vaccination through local health care professionals or medical students, can help support the mass COVID-19 immunization process.

## Figures and Tables

**Figure 1 vaccines-10-00535-f001:**
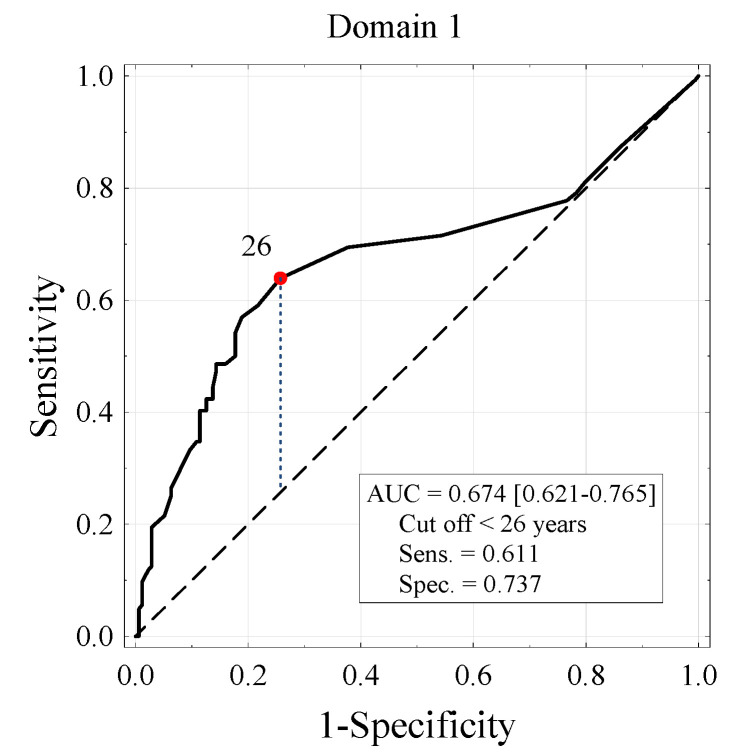
ROC curve for age-based positive domain 1 vaccination estimation. Area under the curve (AUC) and the sensitivity and specificity of the test for the age cutoff <26 years.

**Table 1 vaccines-10-00535-t001:** Sociodemographic characteristics of the study group.

Characteristic (Variable)	*N* (%)
Profession:	
Non-medical staff	8 (1.6%)
Other medical staff	14 (2.8%)
Students	333 (67.1%)
Nurses	108 (21.7%)
Midwives	19 (3.8%)
Paramedics	2 (0.4%)
Doctors	13 (2.6%)
Field of study:	
Nursing	112 (22.6%)
Obstetrics	19 (3.9%)
Dentistry	89 (17.8%)
Pharmacy	60 (12.1%)
Public health	23 (4.7%)
Dietetics	2 (0.5%)
Medicine	176 (35.1%)
Medical analysis	12 (2.5%)
Emergency medicine	4 (0.8%)
Age (years):	
Me (Q1–Q3)	24 (21–28)
Min–Max	18–64

M—mean, SD—standard deviation, Me—median, Q1, Q3—quartile 1 and 3, Min–Max—minimum and maximum values.

**Table 2 vaccines-10-00535-t002:** Questionnaire assessing the attitude of the entire study group towards vaccination against COVID-19.

Domain Number	Questions	The Cronbach’s Alpha Coefficient	Me	Q1–Q3	DomainMe (Q1–Q3)	*p*-Value
Domain 1	1. Do you think that many diseases prevented by vaccination are serious ones, mainly infectious?	0.81	5	5–5	3.4 (2.9–3.9)	<0.001
	2. Do you think that the immunity acquired after contracting the disease is better than after vaccination?		4	3–4		
	3. Do you think that it is better to wait for the next emerging vaccines than to get one of those developed in the first stage?		4	3–5		
	4. Would you make a decision not to vaccinate for reasons other than illness or allergy?		4	3–5		
	5. Would you delay getting vaccinated for reasons other than illness or allergy?		4	3–5		
	6. Do you think that opinions on vaccines are primarily governed by the opinions and benefits of pharmaceutical companies?		4	3–5		
Domain 2	7. Do you think that undergoing the recommended COVID-19 vaccinations is good for your health?	0.841	5	4–5	4.1 (3.6–4.3)	<0.001
	8. Do you think that the timing of mass vaccination against COVID-19 during a period of increased morbidity is reasonable?		5	4–5		
	9. Do you think that vaccination against COVID-19 can protect you from contracting COVID-19?		5	4–5		
	10. Do you think that vaccination against COVID-19 is not needed due to the low probability of getting sick?		5	4–5		
	11. Do you think healthcare professionals should be rewarded for undergoing vaccination against COVID-19?		2	1–3		
	12. Do you think healthcare professionals should be compulsorily vaccinated against COVID-19?		4	3–5		
	13. Do you think mass vaccination against COVID-19 is justified?		5	4–5		
	14. Do you think the current government-organized COVID-19 vaccination schedule and phases are sufficient?		3	2–3		
	15. Do you think the COVID-19 vaccine is effective?		4	3–5		
	16. Do you think that COVID-19 vaccination can prevent people from getting infected due to herd immunity?		5	4–5		
Domain 3	17. Are you worried that side effects may occur after vaccination against COVID-19?	0.834	3	2–4	3.6 (3.0–4.0)	<0.001
	18. Do you think that vaccination against COVID-19 can be dangerous?		4	3–5		
	19. Are you worried that vaccination against COVID-19 may not be effective?		4	2–4		
	20. Are you very concerned about the vaccination against COVID-19 currently being introduced in our country?		4	3–5		
	21. Do you discuss your concerns about vaccinations openly with your GP or other healthcare professional?		2	1–4		
	22. Do you think that COVID-19 vaccination is safe?		2	1–2		
	23. Do you think that the COVID-19 vaccine “causes disease symptoms”?		4	3–5		
	24. Do you think that the COVID-19 vaccine causes immediate, short-term side effects (such as fever or headache)?		3	2–4		
	25. Are you concerned about long-term side effects following the COVID-19 vaccination?		4	2–5		
	26. Do you think that the COVID-19 vaccine causes COVID-19 and weakens the immune system?		5	4–5		
Domain 4	27. Do you think that COVID-19 is a serious threat to health and life?	0.834	1	1–2	1.5 (1.0–2.3)	0.29
	28. Are you afraid of hospitalization due to COVID-19?		2	1–4		
	29. Are you afraid of losing physical or mental fitness due to COVID-19?		1	1–3		
	30. Are you afraid of prolonged loss of professional or social efficiency related to COVID-19?		1	1–2		
	31. Do you think COVID-19 poses a serious threat to the health of others?		1	1–1		
	32. Do you think that the risk of getting COVID-19 is so low that there is no need for vaccination?		1	1–2		

1 (general attitude towards vaccinations) and 2 (positive attitude towards vaccinations); 3 (concerns about the side effects of the COVID-19 vaccine) and 4 (concerns about contracting COVID-19).

**Table 3 vaccines-10-00535-t003:** Attitude towards vaccination against COVID-19 in the studied group depending on occupation, university degree and age.

Characteristic (Variable)	Attitude towards COVID-19 Vaccine Me (Q1–Q3)			
	Domain 1	Domain 2	Domain 3	Domain 4
Gender:				
1. Women (*n* = 255)	3.9 (3.4–4.0)	3.7 (3.3–4.3)	3.3 (2.8–3.6)	1.7 (1.2–2.5)
2. Men (*n* = 242)	3.7 (3.0–4.0)	3.8 (3.5–4.2)	3.2 (2.7–3.7)	1.6 (1.1–2.4)
	*p* = 0.055	*p* = 0.12	*p* = 0.08	*p* = 0.22
Status:	4.2 (3.8–4.7)	4.2 (3.9–4.3)	3.6 (3.2–3.9)	2.5 (1.0–2.2)
1. Medical Students (*n* = 328)	3.7 (3.0–4.3)	3.8 (3.3–4.3)	3.1 (2.6–3.5)	1.3 (1.0–2.1)
2. Health Care Workers (*n* = 169)	*p* < 0.001	*p* < 0.001	*p* < 0.001	*p* = 0.33
Profession:				
1. Non-medical staff (*n* = 8)	3.7 (3.4–3.9)	4.3 (3.9–4.4)	3.4 (3.1–3.9)	1.2 (1.0–1.5)
2. Other medical staff (*n* = 14)	3.4 (2.9–4.3)	4.1 (3.4–4.4)	3.7 (2.4–4.0)	1.8 (1.0–3.0)
3. Students (*n* = 328)	3.6 (3.1–4.0)	4.1 (3.8–4.3)	3.8 (3.2–4.1)	1.7 (1.2–2.5)
4. Nurses (*n* = 108)	3.1 (2.6–3.4)	3.7 (3.3–4.3)	3.0 (2.7–3.5)	1.2 (1.0–2.0)
5. Midwives (*n* = 19)	3.0 (2.4–3.7)	3.4 (3.0–3.8)	3.1 (2.4–3.8)	1.5 (1.0–2.7)
6. Paramedics (*n* = 2)	2.6 (1.6–3.6)	3.3 (2.1–4.5)	3.1 (2.1–4.0)	2.5 (1.0–4.0)
7. Doctors (*n* = 13)	3.9 (3.4–4.0)	4.3 (3.8–4.5)	3.8 (3.0–4.0)	1.7 (1.2–2.5)
	*p* < 0.001	*p* < 0.001	*p* < 0.001	*p* = 0.012
Field of study:				
1. Nursing (*n* = 112)	3.1 (2.6–3.6)	3.7 (3.3–4.3)	3.1 (2.7–3.6)	1.2 (1.0–2.0)
2. Obstetrics (*n* = 19)	3.0 (2.4–3.7)	3.4 (3.0–3.8)	3.1 (2.4–3.8)	1.5 (1.0–2.7)
3. Dentistry (*n* = 89)	3.3 (2.7–4.0)	4.1 (3.6–4.4)	3.7 (2.9–4.0)	1.5 (1.2–2.5)
4. Pharmacy (*n* = 60)	3.6 (2.9–4.0)	4.1 (3.7–4.4)	3.8 (3.3–4.1)	1.8 (1.2–2.8)
5. Public health (*n* = 23)	3.3 (2.9–3.6)	3.9 (3.5–4.3)	3.3 (3.0–3.8)	1.7 (1.2–2.3)
6. Dietetics (*n* = 2)	3.4 (3.0–3.9)	3.9 (3.2–4.5)	3.2 (2.4–3.9)	1.7 (1.0–2.3)
7. Medicine (*n* = 176)	3.6 (3.3–4.0)	4.2 (3.9–4.3)	3.9 (3.3–4.1)	1.6 (1.0–2.3)
8. Medical analysis (*n* = 12)	3.6 (3.3–3.9)	4.2 (3.8–4.4)	3.6 (3.0–3.9)	1.6 (1.2–2.4)
9. Emergency medicine (*n* = 4)	3.6 (2.4–3.9)	3.4 (2.6–3.7)	3.9 (2.9–4.2)	2.3 (1.8–3.3)
	*p* < 0.001	*p* < 0.001	*p* < 0.001	*p* = 0.041
Age (years):				
18–24 (*n* = 208)	3.6 (3.0–4.0)	4.2 (3.7–4.4)	3.7 (3.2–4.0)	1.7 (1.2–2.5)
25–34 (*n* = 90)	3.6 (3.0–4.0)	4.1 (3.6–4.3)	3.7 (2.8–4.1)	1.3 (1.0–2.0)
35–44 (*n* = 33)	3.0 (2.3–3.6)	3.7 (3.0–4.3)	3.4 (2.3–3.8)	1.7 (1.0–2.7)
45–54 (*n* = 52)	3.1 (2.9–3.4)	3.8 (3.5–4.3)	3.1 (2.8–3.4)	1.2 (1.0–1.6)
55–64 (*n* = 14)	2.7 (2.4–3.0)	3.8 (3.3–3.9)	3.1 (2.4–3.4)	4.2 (3.7–4.4)
	*p* < 0.001	*p* = 0.001	*p* < 0.001	*p* = 0.001

1 (general attitudes towards vaccination) and 2 (positive attitudes towards vaccinations); 3 (concerns about the side effects of the COVID-19 vaccine) and 4 (concerns about COVID-19). Me—median, Q1–Q3—interquartile range.

**Table 4 vaccines-10-00535-t004:** Attitudes towards vaccination against COVID-19 and results for domains 1 and 2.

Attitudes towards Vaccination against COVID-19 (Points)	Number (Percentage) of Respondents							
	Domain 1		Domain 2		Domain 3		Domain 4	
	*n*	%	*n*	%	*n*	%	*n*	%
Strongly negative (1)	5	1.0	4	0.8	6	1.2	220	44.2
Rather negative (2)	72	17.7	16	3.2	66	13.5	168	33.5
No opinion (3)	183	37.1	67	13.9	149	30.3	83	17.1
Rather positive (4)	237	47.2	332	67.1	252	50.2	22	4.4
Strongly positive (5)	0	0.0	75	14.9	24	4.8	4	0.8
Me (Q1–Q3)	3 (3–4)		4 (4–4)		4 (3–4)		2 (1–2)	
Min–Max	1–4		1–5		1–5		1–5	

Domain 1—general attitude towards vaccinations; Domain 2—positive attitude towards COVID-19 vaccination; M—mean, SD—standard deviation; Me—median, Q1, Q2—quartile 1 and 2, Min–Max—minimum and maximum values.

**Table 5 vaccines-10-00535-t005:** The values of Spearman’s rank correlation coefficients between age and the assessment of the attitude to vaccination in the four domains of the questionnaire.

	Domine 1 [Pts.]	Domine 2 [Pts.]	Domine 3 [Pts.]	Domine 4 [Pts.]
Age [years]	rho = −0.301 *p* < 0.001	rho = −0.246 *p* < 0.001	rho = −0.176 *p* < 0.01	rho = 0.072 *p* > 0.05

**Table 6 vaccines-10-00535-t006:** Number (proportion) of people in groups differing in vaccination ratio and age, and the results of Chi-square tests of independence and odds ratios.

	Age (Years)	Attitude Against Vaccinations		*p*-Value	OR (95% CI)
		Positive	Negative		
Domain 1	<26	130 (74.3%)	52 (36.1%)	<0.001	5.11 (3.16–8.22)
	≥26	45 (25.7%)	92 (63.9%)		1.00 (ref.)
Domain 2	<26	125 (71.4%)	57 (39.6%)	<0.001	3.82 (2.39–6.09)
	≥26	50 (25.7%)	87 (60.4%)		1.00 (ref.)
Domain 3	<26	34 (82.9%)	148 (53.2%)	<0.001	4.27 (1.83–9.95)
	≥26	7 (17.1%)	130 (46.8%)		1.00 (ref.)
Domain 4	≥24	20 (80.0%)	178 (60.5%)	0.087	2.61 (0.95–7.14)
	<24	5 (20.0%)	116 (39.5%)		1.00 (ref.)

**Table 7 vaccines-10-00535-t007:** Number (proportion) of people in groups differing in vaccination ratio and status, and the results of Chi-square tests of independence and odds ratios (row and adjusted)**.**

	Status	Positive Attitude towards Vaccinations		*p*	OR (95% CI)	OR_adj._ (95% CI)
		Yes	No			
D1	Medical student	120 (68.6%)	43 (29.9%)	<0.001	5.12 (3.17–8.27)	2.58 (1.20–5.58)
	Health professional	55 (31.4%)	101 (70.1%)		1.00 (ref.)	1.00 (ref.)
D2	Medical student	116 (66.3%)	47 (32.6%)	<0.001	4.06 (2.54–6.48)	2.20 (1.03–4.66)
	Health professional	59 (33.7%)	97 (67.4%)		1.00 (ref.)	1.00 (ref.)
D3	Medical student	33 (80.5%)	130 (46.8%)	<0.001	4.70 (2.09–10.5)	3.15 (0.85–11.6)
	Health professional	8 (19.5%)	148 (53.2%)		1.00 (ref.)	1.00 (ref.)
D4	Medical student	8 (32.0%)	155 (52.7%)	0.075	0.42 (0.18–1.01)	0.44 (0.14–1.41)
	Health professional	17 (68.0%)	139 (47.3%)		1.00 (ref.)	1.00 (ref.)

OR_adj._—age-adjusted odds ratio.

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
