# Peer review of "Attitude of Health Care Workers and Medical Students towards Vaccination against COVID-19"

_vaccines, 2022, doi:10.3390/vaccines10040535_

Round 1
Reviewer 1 Report
It's an interesting and important topic, as health workers are very important in increasing awareness and demand among communities towards vaccination against pandemic and prevention.
However, paper needs more up to date references and some revision of text in findings section and Discussion.
Line 40-41: needs rephrasing. “Limiting virus transmission” is an output of preventive measures, not itself a preventive measure.
Line 66: currently 66% of population has received at least one dose of Covid-19 vaccine in Poland, >78% in France. Statement and reference need to be corrected, it’s not smallest number of people.
Line 85: text starting “Moreover,…..” is not clear or incomplete.
Line 296: statement doesn’t match with findings of this study.
Discussion sections needs to be rewritten to analyze findings if this study with other studies. Recommendations needs to be improved too.
Author Response
Line 40-41: needs rephrasing. “Limiting virus transmission” is an output of preventive measures, not itself a preventive measure.
Thank you, the excerpt has been corrected according to the comments.
Line 66: currently 66% of population has received at least one dose of Covid-19 vaccine in Poland, >78% in France. Statement and reference need to be corrected, it’s not smallest number of people.
Thank you, the excerpt has been corrected. The introduction has been modified and condensed
Line 85: text starting “Moreover,…..” is not clear or incomplete.
Thank you, the excerpt has been corrected
Line 296: statement doesn’t match with findings of this study.
Thank you, the excerpt has been corrected. Discussion has been modified
Discussion sections needs to be rewritten to analyze findings if this study with other studies. Recommendations needs to be improved too.
Thank you, the passage has been corrected, the discussion has been expanded.
Reviewer 2 Report
I was invited to revise the paper entitled "Knowledge and attitude of health care workers and medical students towards vaccination against COVID-19". It was a cross-sectional study aimed to assess the attitudes of HCW and medical students towards vaccinations and associated factors.
The topic is interesting but I have some major observations:
- Sample size estimation was lacking;
- Non medical students should be excluded from the analysis;
- the number of HCW is too different compared to students. It can lea to bias;
- Line 132: Kruskall Wallis test is not ANOVA !
- Authors cannot perform a linear regression analysis for discrete variable and for variables with non normal distribution. Authors should use different methods;
- In table 1 authors should divide partecipants by working status. Students are always younger than HCW, with different knowledge towards preventive measures;
- Scores are discrete variables and have to be reported as median and IQR;
- Line 177: no correlation tests were performed;
- Chapter 3.2: Authors should report also the IQR with median results example line 179 "Students had a significantly more positive attitude towards vaccination (domain 1) than nurses (MEDIAN (IQR) vs. MEDIAN(IQR) ; p <0.001);
- Analyses reported in Table 3 should be modified. Authors should consider Students as a single category and comapare it against HCW. Subgroup analysis should be added after the main analysis;
- Authors should add pairwise comparison for KW tests;
- Why Authors used onle age and working status as factors associated to hesitancy?
- In Table 6.7,8,9 Authors have to remove one between age as continouns variable and age category. They cannot be used simoultaneusly;
- Discussion should focus on differences between HCW and students;
- Hesitancy by age category should be better discussed;
- diffecences with previous study should be reported;
- Limitation section was missing.
Author Response
- Sample size estimation was lacking;
It was estimated that a minimum of 62 subjects would be needed to assess the differences between the proportions in the two study groups, given a Type I error of 0.05 and a power of 90%.
- Non medical students should be excluded from the analysis;
- Thank you for your comment. Non-medical students were excluded from the study
- the number of HCW is too different compared to students. It can lead to bias;
- - Thank you for your comment. We have incorporated this suggestion into the study limitation.
- Line 132: Kruskall Wallis test is not ANOVA !
- Thank you for your comment. The erroneous section has been corrected.
- Authors cannot perform a linear regression analysis for discrete variable and for variables with non normal distribution. Authors should use different methods;
- Thank you for your comment. The authors used different statistical methods.
- In table 1 authors should divide partecipants by working status. Students are always younger than HCW, with different knowledge towards preventive measures;
- Thank you for your comment. An analysis of the effect of age on attitudes toward vaccination in the results section was conducted.
- Scores are discrete variables and have to be reported as median and IQR;
- Thank you for your comment. All results are now presented using medians and IQR
- Line 177: no correlation tests were performed;
- Thank you for your comment. The correlation was carried out
- Chapter 3.2: Authors should report also the IQR with median results example line 179 "Students had a significantly more positive attitude towards vaccination (domain 1) than nurses (MEDIAN (IQR) vs. MEDIAN(IQR) ; p <0.001);
- Thank you for your comment. All results are now presented using medians and IQR
- Analyses reported in Table 3 should be modified. Authors should consider Students as a single category and comapare it against HCW. Subgroup analysis should be added after the main analysis;
- Thank you for your comment. The authors chose to present the data in this form because the number of participants among HCWs and students was proportional to the number of people working or practicing at the research center. The university hospitals among which individuals were surveyed are characterized by having twice as many students as medical staff.
- Authors should add pairwise comparison for KW tests;
- Thank you for your comment. The statistical analysis has been revised and modified
- Why Authors used onle age and working status as factors associated to hesitancy?
- Thank you for your comment. Additional factors that were analyzed were job tenure/year of study and gender. However, no significant statistical differences were shown, so the authors chose not to present these results.
- In Table 6.7,8,9 Authors have to remove one between age as continouns variable and age category. They cannot be used simoultaneusly;
- Thank you for your comment. The analyses presented in Tables 6-9 have been replaced by others where the factor that was tested was age
- In table 1 authors should divide partecipants by working status. Students are always younger than HCW, with different knowledge towards preventive measures;
- Thank you for your comment, analysis of age as a factor influencing attitudes toward puppies is presented in the resluts section.
- Discussion should focus on differences between HCW and students;
Thank you for your comment. The discussion has been revised according to the comments.
Hesitancy by age category should be better discussed;
- Thank you for your comment. The discussion has been revised according to the comments.
- diffecences with previous study should be reported;
- Thank you for your comment. The discussion has been revised according to the comments.
- Limitation section was missing.
Thank you for your comment. Limitations have been added.
Reviewer 3 Report
The study entitled “Knowledge and attitude of health care workers and medical students towards vaccination against COVID-19” is a cross-sectional survey aimed to: 1) assess attitude of medical workers and students of medical faculties towards vaccinations; 2) and, to determine factors influencing decisions regarding vaccinations. The authors highlighted the important role of mid-level medical personnel in the prevention and promotion of pro-health behaviors. This is a very important issue from a public health point of view, however, in its current version, the study presents some issues that must be addressed before it could be considered worthy of publication on Vaccines. I pointed below some comments in order to help the authors to improve their manuscript.
- In order for the readers to better understand, you could annex the Questionnaire, in English language, as a supplement file to the publication.
- I suggest that authors revise the manuscript for language usage.
- Introduction
- In the Introduction section there are too much information, that are not relevant to the objectives of the study.
- The aims of this study are not very clear. In the Title they referred to COVID-19 vaccination, in the aims only vaccination in general. Please better clarify this point.
- Methods
- In the Methods section it is not very clear the selection of participants for the study. Please specify how they were selected. There is the need more information on all of these things as they impact interpretation of results.
- Was there any inclusion criteria for participants to be included in the study? If yes, what were they? Was there any exclusion criteria? If yes, what?
- The authors should add the references of the studies used for the construction of the questionnaire.
- Results
- In the title and in the aims of this study, the authors refer to “knowledge” of health care workers and medical students towards vaccination against COVID-19. In the Results section, the authors reported only results on the attitude. Please clarify.
- Statistical analyses are not well described.
- There are too many Tables, but they are not very clear.
- Discussion
- The literature review and background is not wide and comprehensive.
- There are others investigations which could be cited to compare with your results. For examples:
- Della Polla G, Licata F, Angelillo S, Pelullo CP, Bianco A, Angelillo IF. Characteristics of healthcare workers vaccinated against influenza in the era of COVID-19. Vaccines (Basel) 2021;9:695. doi: 10.3390/vaccines9070695.
- Di Giuseppe G, Pelullo CP, Paolantonio A, Della Polla G, Pavia M. Healthcare workers’ willingness to receive influenza vaccination in the context of the COVID-19 pandemic: a survey in Southern Italy. Vaccines. 2021;9:766. doi.org/10.3390/vaccines9070766.
- References
Please make sure you comply with reference style of Vaccines.
Author Response
- In order for the readers to better understand, you could annex the Questionnaire, in English language, as a supplement file to the publication.
Thank you for your comment. The questionnaire was presented in full in Table 2
- I suggest that authors revise the manuscript for language usage.
- Thank you for your comment. The text has been linguistically checked.
- Introduction
- In the Introduction section there are too much information, that are not relevant to the objectives of the study.
Thank you for your comment. The introduction has been revised and shortened.
- The aims of this study are not very clear. In the Title they referred to COVID-19 vaccination, in the aims only vaccination in general. Please better clarify this point.
The purpose was to examine attitudes toward receiving the COVID-19 vaccine, although general attitudes toward vaccination may influence willingness to receive the aforementioned vaccine. Given this, one of the 4 domains of the questionnaire included an assessment of general attitudes toward vaccination.
- Methods
- In the Methods section it is not very clear the selection of participants for the study. Please specify how they were selected. There is the need more information on all of these things as they impact interpretation of results.
The directors of each medical center sent invitations to participate in the study to the participants' individual email addresses.
- Was there any inclusion criteria for participants to be included in the study? If yes, what were they? Was there any exclusion criteria? If yes, what?
Eligibility and exclusion criteria are provided in the methods
- The authors should add the references of the studies used for the construction of the questionnaire.
General attitudes toward vaccination are related to attitudes toward accepting other vaccines including COVID-19. Attitudes toward vaccination are also influenced by fear of negative effects of the vaccine and fear of contracting COVID-19.
- Results
- In the title and in the aims of this study, the authors refer to “knowledge” of health care workers and medical students towards vaccination against COVID-19. In the Results section, the authors reported only results on the attitude. Please clarify.
Thank you very much for your attention. The purpose of the study changed during the study and the final analysis focused on attitudes toward vaccination. The title of the paper was modified.
- Statistical analyses are not well described.
Thank you for your comment. The statistical methods are better described.
- There are too many Tables, but they are not very clear.
Thank you for your comment. The number of tables has been reduced.
- Discussion
- The literature review and background is not wide and comprehensive.
Thank you for your comment. Background has been corrected.
- There are others investigations which could be cited to compare with your results. For examples:
- Della Polla G, Licata F, Angelillo S, Pelullo CP, Bianco A, Angelillo IF. Characteristics of healthcare workers vaccinated against influenza in the era of COVID-19. Vaccines (Basel) 2021;9:695. doi: 10.3390/vaccines9070695.
- Di Giuseppe G, Pelullo CP, Paolantonio A, Della Polla G, Pavia M. Healthcare workers’ willingness to receive influenza vaccination in the context of the COVID-19 pandemic: a survey in Southern Italy. Vaccines. 2021;9:766. doi.org/10.3390/vaccines9070766.
Thank you for your comment. Both items were cited in the discussion.
- References
Please make sure you comply with reference style of Vaccines.
Thank you for your comment. References were checked with reference style of Vaccines.
Reviewer 4 Report
This is an interesting cross-sectional study on the attitudes of HCWs towards COVID 19 vaccination.
The topic is of importance and the manuscript is well written. I believe there are some points which deserves further clarifications be the authors and this will improve further the quality of this manuscript.
- Abstract. Please provide the response rate.
- Introduction. My general impression is that the introduction is too long and needs shortening. The authors should avoid repetition of well known details related to COVID 19 pandemic and focus on vaccine hesitancy among HCWs.
- Results. I missed the response rate. In addition, given that reliability (internal consistency) analysis was performed the authors should report the exact a Cronbach value for each domain and for the questionnaire as a whole. A score was used in the presentations of the results. This is acceptable; however, it would be very interesting for the reader to see the descriptive analysis of the attitudes by the use of percentages. Lastly, there are 9 tables presenting the results. This is not practical for the reader and the authors could reduce the number of tables and focus on the multivariate analysis results.
- Discussion. In the first paragraph the authors discussed the issue of herd immunity. However, nowadays, it is difficult or even impossible to define cut-off levels of herd immunity, because the vaccines mainly protect against severe disease and death. I am not clear on the role of nursing profession. The authors should elaborate further on this important point. The limitations of the study deserve further discussion.
- References. The following references may be of use: Papagiannis et al. Acceptability of COVID 19 vaccination among Greek health professionals. Vaccines 2021, 9, 200
- Marinos G et al. Reported vaccination coverage against COVID 19 and associated factors.... Vaccines 2021, 9, 1154
Author Response
- Abstract. Please provide the response rate.
Thank you for your comment. Introduced response rate information in the results section.
- Introduction. My general impression is that the introduction is too long and needs shortening. The authors should avoid repetition of well known details related to COVID 19 pandemic and focus on vaccine hesitancy among HCWs.
Thank you for your comment. The introduction has been shortened. Information on vaccine hesitancy among HCWs has been highlighted. Moreover, additional analyses related to age and attitudes toward vaccination were introduced
- Results. I missed the response rate. In addition, given that reliability (internal consistency) analysis was performed the authors should report the exact a Cronbach value for each domain and for the questionnaire as a whole. A score was used in the presentations of the results. This is acceptable; however, it would be very interesting for the reader to see the descriptive analysis of the attitudes by the use of percentages. Lastly, there are 9 tables presenting the results. This is not practical for the reader and the authors could reduce the number of tables and focus on the multivariate analysis results.
Thank you for your comment. Information on Cronbach value has been introduced. Number of tables has been reduced. Response rate information has been added to the results section. Percentages of attitudes toward vaccination are present in Table 4.
- Discussion. In the first paragraph the authors discussed the issue of herd immunity. However, nowadays, it is difficult or even impossible to define cut-off levels of herd immunity, because the vaccines mainly protect against severe disease and death. I am not clear on the role of nursing profession. The authors should elaborate further on this important point. The limitations of the study deserve further discussion.
Thank you for your comment. The section on collective resilience has been removed and the discussion has been more focused on outcomes.
- References. The following references may be of use: Papagiannis et al. Acceptability of COVID 19 vaccination among Greek health professionals. Vaccines 2021, 9, 200
Marinos G et al. Reported vaccination coverage against COVID 19 and associated factors.... Vaccines 2021, 9, 1154
Thank you for your comment. This article was cited in the discussion.
Round 2
Reviewer 1 Report
Thaks for addressing all comments
Author Response
Dear Reviewer,
We would like to thank the Editor and the Reviewers for positive evaluations.
With kind regards, awaiting your respected decision,
Kathie Sarzyńska
Reviewer 2 Report
I was invited to review the revised version of the paper entitled "Knowledge and attitude of health care workers and medical students towards vaccination against COVID-19".
Authors tried to address all point raised during the first review round but I have some observation:
- Comments on table 3 were ignored;
- Authors stated " Additional factors that were analyzed were job tenure/year of study and gender. However, no significant statistical differences were shown, so the authors chose not to present these results". Authors have to show these results as Supplementary material. In addition, no statement about this point were added in the main paper;
- I have many concern about sample size calculation: Authors stated that "It was estimated that a minimum of 62 subjects would be needed to assess the differences between the proportions in the two study groups, given a Type I error of 0.05 and a power of 90%". Which calculation was performed? Which was the considered population ? Which kind of difference was tested?
- Differences in study group was not addressed but only a statement in limitation section was added;
- Authors replaced linear regression with correlation analysis. Why? It is unclear;
- I highlighted the impact of working status on the knowledge of participants but Authors performed an analysis by age. Why?
Author Response
Dear Reviewer,
We would like to thank the Editor and the Reviewers for positive evaluations.
The final version of the manuscript text includes all necessary modifications and improvements as indicated in the table below. We are at your disposal for any further modifications that you find advisable.
We very much hope that our carefully prepared response appears comprehensive and proves helpful in obtaining a positive final decision accepting our paper for publication in your prestigious journal.
With kind regards, awaiting your respected decision,
Kathie Sarzyńska

Reviewer 4 Report
All my concerns have been adequately addressed by the authors.
Author Response

(The authors gave the same response as above.)

Round 3
Reviewer 2 Report
Authors now have addrerssed the great part of points raised. I'm still perplexed about sample size calculation. It is unclear which is the main outcome used to estimate the sample size and I'm not sure it is correct. Probably it was post-estimated. In addition some explanation about methodology remain unsolved.
Author Response
Thank you for your comment.
When comparing the attitudes of respondents towards vaccination, contingency tables and Chi-square tests of independence were used. The fractions in two groups were compared. It was assumed that the fraction differences will be significant when they amount to at least 20%. Assuming Type I error alpha = 0.05 and 1-beta test power = 0.80, the minimum size in each group should be N1 = N2 = 76.
The numbers in the students and health care workers groups were greater than the minimum values of N1 and N2 = 76.
We have included this information in the methods section.
